

Role of Microbial Communities in the Weathering and Stalactite Formation in Karst
Topography
Tung-Yi Huang [a], Bing-Mu Hsu [abc #], Cheng-Wei Fan [a*], Hsin-Chi Tsai[de*], Chien-Yi Tung[fg],
Jung-Sheng Chen[a]
[a] Department of Earth and Environmental Sciences, National Chung Cheng University
address: 168 University Road, Minhsiung Township, Chiayi County 62102, Taiwan
[b] Center for Innovative on Aging Society, National Chung Cheng University
address: 168 University Road, Minhsiung Township, Chiayi County 62102, Taiwan
[c] Center for Nano Bio-detection, National Chung Cheng University
address: 168 University Road, Minhsiung Township, Chiayi County 62102, Taiwan
[d] Department of Psychiatry, School of Medicine, Tzu Chi University, Hualien, Taiwan
address: 701, Sec. 3, Jhongyang Road. Hualien 97004, Taiwan
[e] Department of Psychiatry, Tzu-Chi General Hospital, Hualien, Taiwan
address: 707, Sec. 3, Jhongyang Road. Hualien 97004, Taiwan
[f] Caner Progression Research Center, National Yang-Ming University
address: 155, Li-Nong St., Sec.2, Beitou, Taipei, 11221, Taiwan
[g] Institute of Microbiology and Immunology, National Yang-Ming University
address: 155, Li-Nong St., Sec.2, Beitou, Taipei, 11221, Taiwan

[*] Equal contribution to first author
[#] Corresponding author
E-mail: bmhsu@ccu.edu.tw
TEL: 88652720411 #66218




Abstract
This study investigated the long-term effect of environmental physical factors on the
relative abundance of bacteria and the consequential landscape evolution in karst topography,
focusing mainly on the effects of limestone weathering and calcite precipitation. The Narrow-
Sky located in the upper part of Takangshan is a small gulch of Pleistocene coralline limestone
formation in southern Taiwan. The landscapes were different in the karst walls between the
opening and the inner of gulch due to the variation of physical parameters such as sunlight
penetration, humidity, and temperature. A metagenomic approach was used out to determine
the relationship of microbial community structures on the landscapes in various habitats around
the gulch, namely on the inner and outer limestone wall, the water collected from speleothems
surface, and the ground soil at the outer wall. The total organic carbon content was measured
in solid samples to evaluate the biomass of the habitats. Our results showed that the biomass
of habitats in the opening of the gulch was two times higher than the that inside where light
penetration was lower. We also found that speleothems only occurred at the inner wall inside
the gulch, where the environment exhibited water drips running through the surface of
speleothems and less light penetration. The metagenomics in each habitat were surveyed to
measure the sequence similarity of operational taxonomic units relative to urease-producing
bacteria and weathering-associated bacteria available in the National Center for Biotechnology
Information database. Our data revealed that the metagenomics of the inner wall and water
samples exhibited more sequences that were similar to those of urease-producing bacteria,
whereas the outer wall showed more sequences that were similar to those of weathering-
associated bacteria, suggesting that bacteria facilitated the formation of limestone weathering
and calcite precipitation for various habitats. This study revealed the pivotal roles of
microorganisms in governing the geological evolution on the limestone landscape.





### Introduction

Weathering and calcite precipitation are two opposite activities that affect the dynamic changes of the karst landscape. Although weathering and calcite precipitation can occur in abiotic conditions, several lines of evidence from cave studies or laboratory data have shown that microorganisms can accelerate the reactions that promote the formation of calcium carbonate and the breakdown of calcite in situ (Gat et al., 2014;Sulu-Gambari, 2011;Castanier et al., 1999;Lian et al., 2008;Jones, 2017). During the breakdown of carbonate rocks, microbial colonies build up on rock surfaces, resulting in rock decomposition by acidification and moisturization onto the surfaces (Wu et al., 2017;Hutchens, 2009;Uroz et al., 2009). The obtainment of nutrients from the rock surface further promotes the release of organic ligands, which in turn facilitate the release of mineral elements, thus creating a positive feedback loop (Lian et al., 2008;Uroz et al., 2007). Many studies have documented that the mineral dissolution of rocks in a flow-through system was higher in the presence of microorganisms, whereas the dissolution was enhanced in the groups of surface-attached microorganisms, especially when compared with the unattached ones (Ahmed and Holmström, 2015;Seiffert et al., 2014;Jacobson and Wu, 2009). Many bacterial strains have been reported to have the ability to adhere to rock surfaces and establish the weatherability (Sulu-Gambari, 2011). For example, *Shewanella oneidensis* can recognize silicate and oxide mineral surfaces and cause further weathering associated reactions (Lower et al., 2001). To date, many studies have categorized the bacteria of weatherability (Uroz et al., 2009;Sulu-Gambari, 2011;Lian et al., 2008).

Many bacteria can induce the biomineralization processes of calcium carbonate precipitation that render the formation of stalactite. The microbial-induced reaction is mainly carried out by urease-producing bacteria in the presence of ammonium ions in the alkaline environment. The identified urease-producing bacteria have been investigated extensively (Anbu et al., 2016;Wei et al., 2015;Ercole et al., 2001;Jones, 2017;Animesh and Ramkrishnan, 2016;Abo-El-Enein et al., 2012). The microbial communities of karst habitats are diverse, and their components largely depend on the locations and composition of limestone (Barton and Northup, 2007;Ortiz et al., 2014;Tomczyk-Żak and Zielenkiewicz, 2015;Engel, 2010). Temperature, light intensity, and light penetration are important parameters that control the developing orientation of microbial communities. Researchers have shown that microorganisms, operating together with the local environmental conditions, play important roles in remodeling the landscapes of karst (Castanier et al., 1999;Mortensen et al.,



2011;Qabany et al., 2012;Anbu et al., 2016). However, how physical factors affect microbial
communities and the consequent geological changes remains unknown. Determining how
microbial relative abundance shifted in response to changes in environmental factors and the
consequent geological evolution can enable us to better understand the effect of
microorganisms on the dynamic alterations of karst landscapes.
The most abundant limestone is found in warm and humid regions. Because of its porous
and loose property, limestone can be easily infiltrated by rainfall or groundwater to form
trenches, shallow concavity, or clefts. Limestone landscapes in Taiwan are scattered all over
the island and can be found in the Hengchun Peninsula, east coastal areas, central range, and
southwest of Taiwan. The tectonic studies of Takangshan revealed that the upthrow consists of
large lenses of Pleistocene anticline, 4 km in length and 2 km in width (Lacombe et al., 1997;
M.L. Hsieh and Knuepfer, 2001). The crest of Takangshan is covered with coral reef limestones
(with an average thickness of 40 m), which are interbedded in clastic layers. On top of the hills,
expanding vegetation coverages, coupled with erosion soil, are commonly observed in most of
these limestone landscapes. The Narrow-Sky is a nickname for a mountain crack located at a
limestone hill in the Tainliao district of Taiwan. The dimensions (length, width, and height) of
the gulch is approximately 100 m, 2 m, and 12 m, respectively. Because of vegetation
coverages and its topographic features, the exposure of sunlight at different spots inside the
gulch is different. For example, sunlight can penetrate the limestone wall of the opening
through the vegetation coverage, while it is relatively dim inside the path of the gulch.
Moreover, moisture and temperature are also different between the opening and the center of
the gulch. The most tangible difference between the inner and outer wall of the gulch is the
formation of speleothems, which are plentiful in the inner gulch and are nonexistent in the outer
section. Because microbial communities are sensitive to changes in environmental physical
factors, the microbial composition in different locations may have adapted to the environment
according to physical factors, which may play a role in reshaping the gulch scenery.
In this study, we investigated the effect of physical factors on microbial communities in the
limestone landscape. With the recent advent of next-generation sequencing (NGS) platform
and computational methods, we could conduct genome studies on microbes to determine the
relationship between environmental factors in their habitats, such as sunlight penetration during
daytime, humidity, and pH and the relative abundance of microbes. We collected samples from
limestone walls at the opening and inside the gulch, from water dripping at the inner limestone



wall, and from the soil of the outer weathered limestone at the gulch opening: we collected
these samples to extract DNA. Genomic DNA extracted from these samples was further
subjected to the PCR amplification of 16S rRNA gene sequences by using the Illumina's MiSeq
system. Bioinformatics tools were employed to explore DNA reads in operational taxonomic
units (OTUs). The DNA sequence in each OTU was blasted with the sequences which is current
weathering bacteria and urease-producing bacteria available in the National Center for
Biotechnology Information (NCBI) database.



## 2. Materials and methods

### 2.1 Sample Site Description and the Collection of Samples and Physical Parameters

We collected samples from the limestone gulch of Tainliao (120º 21'19.1" E, 22º 51' 00.7" N): the location is illustrated in Figure 1. For collecting the microorganisms in the surface of limestone walls, sterile cotton swabs were used to wipe the surface areas of sampled spots. The samples were collected in a tube and sent to the laboratory to measure total organic carbon and extract DNA for subsequent metagenomics studies. The physical factors in sampling spots including illumination, temperature of the air or soil, humidity, and pH of soil were recorded.

### 2.2 DNA Extraction and PCR for Metagenomics Analysis

The procedure modified from kit of Genomic DNA from soil (Macherey-Nagel) was used to extract bacterial DNA from limestone samples. The detailed procedure was described in detail in our previous study (Huang et al., 2018). In short, DNA in a bulk of soil fraction was isolated and eluted for the PCR amplification of 16S rRNA gene sequences at V3-V4 regions by using Illumina's MiSeq system to create paired-end sequencing data. The target sequence was amplified through PCR by using mixed forward and reverse primers. After separation through electrophoresis in agarose gel, PCR products with expected sizes were harvested.

### 2.3 Metagenomics Library Construction and Analysis

The Illumina Nextera XT index kit was used in the second-stage PCR for the addition of the index. The raw data of forward and reverse reads were aligned using CLC bio's analysis platform (Genomic Workbench v.8.5) with Q20 as a threshold to generate output fasta files. Fasta files were further processed using the sequence analysis tool USEARCH. All sequence fills were merged together, followed by removing duplicates and clustering sequences into OTUs at 97% pairwise identity with the minimum cluster size being set at 2 to construct an OTU-reference library. A comparison between samples and the reference library at a level of 97% sequence identity was made to yield an OTU table, and the number of reads in each OTU was revealed. A 16s UTAX reference database was employed for the assignment of taxonomy for query sequences in the OTU-reference library. We analyzed each habitat by aligning the data, relative abundance, and biodiversity with a heatmap and principal coordinate analysis (PCoA).



### 2.4 Functional Bacteria Analysis

To investigate urease-producing bacteria and weathering-associated bacteria in each habitat, a bioinformatics approach was used to find functional bacteria based on the similarity of DNA sequences. In this method, tables of urease-producing bacteria and weathering bacteria—including bacteria for surface recognition, surface attachment, and mineral dissolutions—were selected from previously published papers in which their corresponding 16S DNA sequences were downloaded from the NCBI database, as shown in Supplementary Tables I and II. The DNA sequence tables were used as references to construct phylogenetic trees by employing the Molecular Evolutionary Genetics Analysis 7 (MEGA 7) program with the setting of parsimony, neighbor-joining, and maximum likelihood analyses. The similarity between adjacent pairs of OTU sequences and reference sequences was tested using the NCBI nucleotide BLAST program. DNA sequences with a similarity of more than 95% were defined as urease-producing bacterial lineages or weathering-associated bacterial lineages of corresponding bacteria, and their read numbers were manually selected to calculate their populations.



## 3. Results

### 3.1 General Description of Environmental Factors

The location of the limestone gulch in Tainliao and the path to the mountain gulch are demonstrated in Figure 1. The left panel of the figure shows the locations of Tainliao, and the right panel describes in detail where soil and karst samples were collected. The water samples were collected from the drippings of stalactites in the gulch. Various environmental parameters were assessed, namely, illumination, temperature, humidity in air, humidity in soil, and pH in soil. The illumination in the gulch was relatively low all year around, ranging from approximately 20 to 600 Lux in a location where reflected light is available, and ranging from approximately 5 to 70 Lux on the wall when measured from 9 AM to 6 PM on a shiny summer day. The illumination at the opening of the gulch ranged from approximately 100 to 800 Lux at a brighter location, but it ranged from approximately 60 to 650 (Lux) on the limestone wall. On the same day, the illumination at an open space around the gulch was approximately 8,000 Lux, 150,000 Lux, 85,000 Lux, and 4,000 Lux at 9 AM, 12 noon, 3 PM, and 6 PM, respectively. The temperature in the inner of the gulch was 2°C – 4°C lower than the temperature at the opening of the gulch. The humidity in the soil versus air was 100% versus 70% ± 5% at the inner gulch and 37.5% ± 22% versus 60% ± 5% at the opening. The pH of the soil was approximately 4.4 – 5 at the inner gulch and approximately 6.2 – 6.6 at the opening. The total organic carbon content in the inner karst wall, the outer karst wall, and the soil of the outer ground was 3.9% ± 0.2%, 7.7% ± 1%, and 9.1% ± 0.5%, respectively. In short, the inner gulch was a zone of relatively lower light penetration compared with the opening of the gulch. The humidity in the air was similar in the inner and outer gulch. The relative light penetration in the outer gulch may affect the level of local biosynthesis, resulting in a higher total organic carbon content in the areas.

### 3.2 The Microbial Community Structure in Various Karst Habitats Based on Results of the NGS Platform

The metagenomic sequence data from different habitats, namely from the outer soil, the outer karst, the inner karst, and from dripping water, contained a similar level of assembled reads that were clustered into OTUs, revealing a high variety in numbers, as shown in Table 1. The average read number was more than 400,000. The sample from the soil of the outer gulch had the highest OTUs, whereas the water sample had the lowest OTUs. Because the



Shannon index of the sample from the outer gulch was the highest, the effective number of
species was also the highest.

Figure 2 shows the relative abundance of OTUs in different habitats, which contained 22
phyla in total. Our results revealed that the soil sample from the outer gulch had the highest
alpha diversity, whereas the water sample had the lowest alpha diversity. Four major phyla,
namely Proteobacteria, Acidobacteria, Actinobacteria, and Cyanobacteria, accounted for 80%
of total microbial species in all the groups. Moreover, Cyanobacteria was present in both
limestone walls and was absent in water and soil habitats. Although Actinobacteria can be
found in freshwater habitats, our results revealed that they accounted for only <0.4% of the
relative abundance in the karst dripping water. The right panel of Figure 3 shows the heatmap
of OTUs in various habitats. Our data revealed that microorganisms around the gulch were
considerably diverse, and the OTU pattern of the water sample was markedly different from
those of other samples. The habitats of the outer gulch, outer soil, and outer karst wall more
closely resembled each other than the outer and inner karst wall. These findings suggest that
the effect of light penetration and moisture overwhelmed the effects of chemical
compositions in the karst walls. The right panel of Figure 3 shows the PCoA distribution of
dominated OTUs in the environment, indicating that the distribution of bacteria in the karst
gulch was considerably diverse. Many unique OTUs were present in the water habitat (blue
square). Although the number of each OTU between the samples of the outer karst wall and
soil more closely resembled (Figure 3, right panel), the PCoA showed that the distribution of
many dominant OTUs in inner (orange cross) and outer (green cross) karst walls was
adjacent to each other, suggesting that the sequences of dominant species in these two
habitats were similar to each other.

**3.3 Distribution of Weathering Bacteria in the Karst Gulch**
Table 2 shows the OTUs of habitats with sequences that had >95% similarity to reference
sequences of weathering bacteria. Supplementary Table I shows bacteria collected in previous
studies, indicating that they were capable of promoting the functions of weathering in rocks.
Most of the weathering-associated bacteria in habitats belonged to the phyla Proteobacteria.
Although 220 species of weathering-associated bacteria were used as references, only <10%
of them showed similarity to OTU sequences in the karst habitats in this study. The relative
abundance of weathering-associated bacteria in each habitat is shown in Figure 4. The sample
from the inner karst gulch contains the last portions of bacteria relative to weathering-





associated bacteria. The dominant genus in the rock and soil of weathering-associated bacteria
in the karst gulch was *Sphingomonas* whereas *Noviherbaspirillum* was the unique genus in the
water sample. The existence of weathering-associated bacteria in water indicated that water
plays a role in mediating the propagation of weathering bacteria (Ahmed and Holmström,
2015). Because studies have revealed that microorganisms suspended in liquid can still lead to
the dissolution of elements from rocks, *Noviherbaspirillum* can facilitate the weathering
process of the inner karst wall. However, the relative impact of these bacteria on the weathering
process remains unclear.

**3.4 Distribution of Urease Producing Bacteria in the Karst Gulch**
Table 3 shows the OTUs of habitats with sequences that had >95% similarity to reference
sequences of urease-producing bacteria, which by definition are microbes that can synthesize
enzymes for urea hydrolysis, resulting in subsequent biocalcification in the presence of
calcium ions. The reference table of urease-producing bacteria is shown in supplementary
Table II. The relative abundance of urease-producing bacteria in each habitat is shown in
Figure 5. Urease-producing bacterial lineages in the inner karst wall were related to *Bacillus*
*megaterium*, *B. subtilis*, and *B. mycoides*. In water samples, urease-producing bacterial
lineages were closely related to *B. megaterium* and *Halomonas denitrificans*. Urease-
producing bacteria in both the inner karst wall and dripping water can contribute to the
progress of biocalcification on the inner karst wall. Although the total organic carbon content
on the outer karst wall was two times higher than on the inner karst wall, the numbers of
urease-producing bacterial lineages might only exist in marginal amounts on the outer karst
wall due to the extremely low portion of relative abundance of the bacteria (0.003%, Figure
5). The relative abundance of urease-producing bacterial lineages on the inner wall (both in
water and the karst wall) was approximately 200 times higher than that on the outer karst
wall. The high portion in relative abundance of urease-producing bacteria on the inner wall
indicated that a persistent stalactite formation occurs on the habitat, which is consistent with
its ecological landscape.






## 4. Discussions

Understanding the microbial diversity in the karst landscape provides insights into how bacteria survive in extreme environments and the consequence of geological evolution after their interaction. Many studies focusing on the abundance of microorganisms in karst caves have showed a large microbial diversity in limestone landscapes (Engel, 2010; Ortiz et al., 2014; Ortiz et al., 2013; Tomczyk-Żak and Zielenkiewicz, 2015; Zepeda Mendoza et al., 2016). Most of these studies have confirmed that Actinobacteria and Proteobacteria were the dominant species in karst samples. Our study results revealed that total OTUs distributed in the phyla of Actinobacteria, Proteobacteria, Cyanobacteria, and Acidobacteria in karst habitats were approximately 3500, suggesting the extreme diversity of microorganisms in karst landscapes in our studied site. The bacterial communities from different geological areas exhibited regional difference. For example, the majority of bacterial phyla in karst soil in Guizhou China were Proteobacteria, Actinobacteria, Acidobacteria, and Planctomycetes (Zhou et al., 2009). Our data of karst soil revealed that this habitat exhibited the highest microbial diversity. We posit that weathering bacteria present in the outer karst wall and soil contribute to the nutrient level of the soil, causing a higher total organic carbon content and Shannon index of the soil habitat. Our study indicated that light penetration, together with other physical parameters, specify the development of particular microbial communities as showed in Figure 3. In the long run, the subtle changes of the composition of microbial communities alter the geochemical reactions, rendering the variation of karst landscapes.

With the application of the NGS platform for acquiring metagenomic data in various karst habitats, we could examine the effects of physical parameters on the evolution of microbial communities and the consequential changes in the microenvironment. To make the most of the metagenomic data, we used the sequence similarity tool, BLAST, to determine the likeness of representative DNA sequences of OTUs compared with the functional bacteria available in the NCBI database. Although the relative abundance in the phylum levels of karst habitats was similar, the compositions of functional bacteria tested in each habitat were substantially different. We set the cutting point of similarity at 95% to compare functional bacteria in various habitats, which is approximately the level of the genus. However, it is still under debate whether the 95% cutoff in the DNA sequence similarity is a proper setpoint to cluster a category of functional bacteria. Based on this calculus, our data revealed a large difference in the final results, as shown in Figures 4 and 5, suggesting that a considerable difference exists in the



relative abundance of functional bacteria in different habitats. Further confirmation of specific
functional bacteria in various habitats can be achieved through molecular cytogenetic
techniques, such as fluorescence in situ hybridization.

We hypothesized that the two primary activities of karst landscapes, namely weathering
and stalactite formation, might affect the dynamic changes and geographic evolution of karst
walls. Functional bacteria associated with these activities were analyzed based on the NGS
platform. Our data revealed that a drastic shift in key microorganisms, weathering bacteria and
urease-producing bacteria, occurred in the habitats of various physical parameters, suggesting
that these parameters play a role in the initiation of different paths in geological evolution. The
differences in functional bacterial compositions in various habitats supported the fact that the
speleothem formation occurred primarily in the inner karst wall in the gulch, suggesting
physical conditions in the inner karst wall favor the growth of urease-producing bacteria and
promote calcite precipitation. Studies on Cyanobacteria and calcium precipitation have shown
that microorganisms may highly enhance the precipitation of CaCO3 minerals in hot spring
water (J. Kaźmierczak et al., 1996; Kawano and Obokata, 2007). In this study, the relative
abundance of Cyanobacteria in the inner karst wall was twice as large as the relative abundance
of Cyanobacteria in the outer karst wall, suggesting that the environment of the inner karst is
favorable for the development of Cyanobacteria and the consequential mineral precipitation.

Biocalcification has been widely applied in the ecosystem for many purposes, including
land consolidation, groundwater control, crack remediation, and immobilization of toxic metals
(Anbu et al., 2016; Kumari et al., 2016; S. Animesh and Ramkrishnan, 2016; S.A. Abo-El-
Enein et al., 2012; Uroz et al., 2007). Various bacteria, shown in supplementary Table II,
effectively produce urease, resulting in the precipitation of calcite. Although many
environmental factors that affect the growth conditions of urease-producing bacteria have been
tested, none of the previous studies have investigated the effects of sunlight penetration on the
natural selection of bacterial development. In the study of calcifying bacteria in the Stiffe cave,
*Bacillus* and *Arthrobacter* were isolated from natural habitats, which might have contributed
to speleothem formation. In this study, several distinct features were found from the data of the
NGS platform and the analysis of total organic carbon. First, we found that *B. megaterium* and
*H. denitrificans* were the predominant species among calcifying bacteria. Second, urease-
producing bacteria were dominant in the inner karst wall. Finally, urease-producing bacterial
lineages were also present in the dripping water of the inner wall, which possessed different



species of urease-producing bacteria. Most importantly, the interface between water dripping and the inner karst wall was subjected to the biocalcification effects of both urease-producing bacteria. Our data suggests that bacteria in the water drips of the inner karst wall play an important role in facilitating speleothem formation.

Remarkably, habitats with a lower relative abundance of urease-producing bacteria showed a higher value in relative abundance of weathering bacteria. Meanwhile, the TOC was higher in samples at the gulch opening compared with the sample in the inner wall. We concluded that sunlight and nutrient levels may be two factors affecting TOC in these habitats. Sunlight is an important source providing energy for the accumulation of biomass. Light penetration provided a discriminatory growth condition to heterotrophic microorganisms on habitats in inner and outer walls. In the gulch, more than 90% of the luminance from sunlight was filtered out by the vegetation coverage at the opening of the gulch, and the karst structure of the steep wall further filtered off 0% to 85% of light penetration inside the gulch, depending on the angle of the sun and the horizon, which affects the photosynthesis reaction in these areas. We also noticed that the effective number of species in the soil increased drastically, suggesting that an elevation of mineral nutrients, one important consequence of weathering effect on rocks, caused by the weathering process could provide a favorable growth condition for many other bacteria in the soil. Previous studies have documented how the dissolution of calcite can be enhanced in the presence of heterotrophic microorganisms (Jacobson and Wu, 2009). Our data revealed that the composition of Acidobacteria increased in the habitat of soil, which is consistent with that of a previous study (Zhou et al., 2009). We propose that light penetration plays a pivotal role in natural selection to promote the growth of weathering-associated bacteria, which in turn increase the nutrient level in situ and favor the development of microorganisms.

**Conclusion**

Given an example of the karst landscape, we provided evidence regarding how physical parameters change the microbial community and the consequential landscape evolution. Furthermore, we showed that light penetration regulates the microbial population, leading to the breakdown of calcite, whereas the chemical composition of limestone might deliver certain conditions that limit the growth of bacterial species. These factors, namely light penetration, water dripping, moisture, the chemical composition of karst, and selected bacteria that are intertwined, shape the weathering process and stalactite formation. The natural selections of bacteria were achieved by the preferential growth of two bacterial groups: urease-producing





bacteria in the inner karst wall and weathering bacteria in the outer karst wall. Our data reveals
a causal relationship between environmental factors that contribute to the remodeling of the
topography and are mediated by microorganisms. To the best of our knowledge, this study is
the first to address the distinct role of bacteria in the water dripping of karst in biocalcification
and the effect of light penetration in the microenvironment on the colony selection of microbial
communities.



Figures and tables

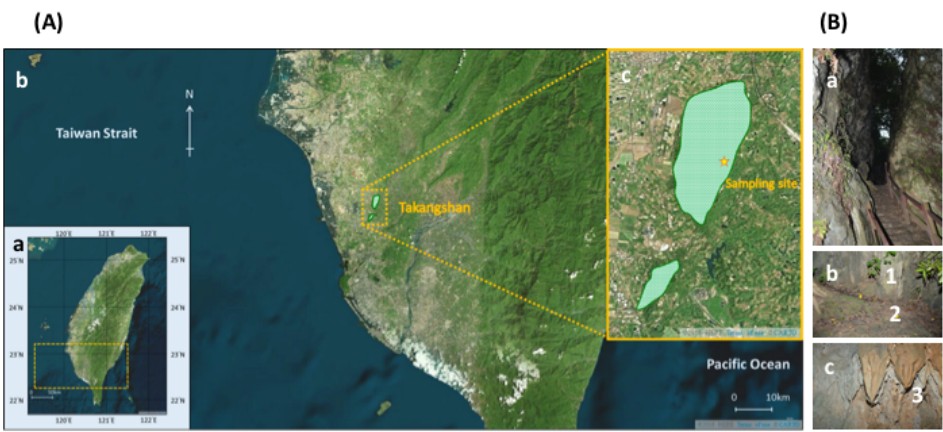


Figure 1. The Location of the Limestone Gulch
Figure (A) shows a series maps of increased scale pointing to the limestone gulch. Figure (B)
shows the view of the gulch. Both sides of the path opening have stairs leading to the center
of the gulch. We took samples from outer limestone wall of the gulch (OL, Figure b1), the
soil on the outer ground (OS, Figure b2), the inner limestone wall of the gulch (IL, Figure
c3), and water dripping from the wall (WA). The inner and outer limestone walls exhibit
distinct landscapes in the stalactite formation.




|  | OS | OL | IL | WA |
|---|---|---|---|---|
| TOC | 9.0±0.4 (%) | 7.7±0.9 (%) | 3.9±0.1 (%) | - |
| Reads | 296,726 | 433,210 | 477,281 | 448,446 |
| OTUs | 2470 | 831 | 1899 | 467 |
| Phyla | 20 | 15 | 19 | 18 |
| Shannon – Wiener index | 6.2 | 4.1 | 3.9 | 3.6 |
| Effective number of species | 511 | 61 | 51 | 36 |





Table 1. The results of Total Organic Carbon, the Basic Information from the NGS Platform,
and the Bacterial Biodiversity in 4 Different Habitats
The symbols of OS, OL, IL, and WA represent sample sites of outer ground, outer limestone
wall, inner limestone wall and water, respectively.




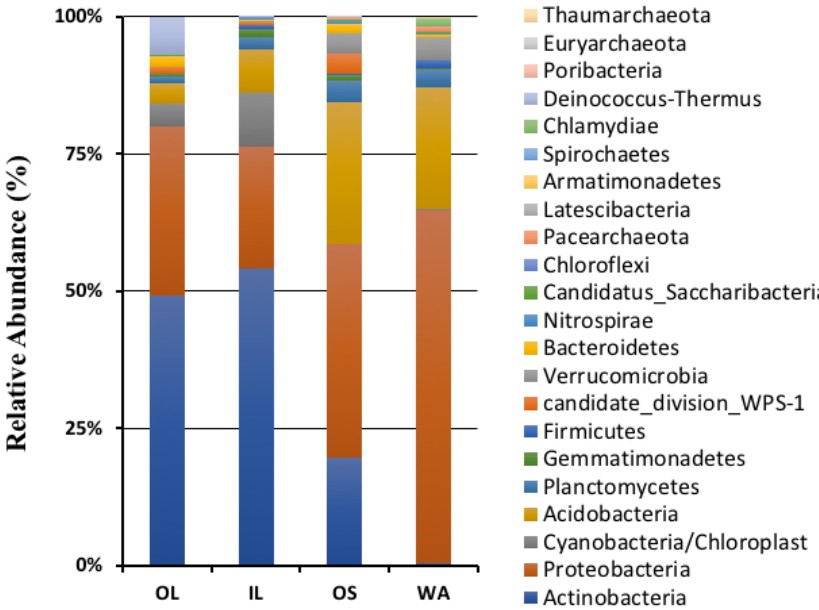


Figure 2. The Relative Abundance of 4 Various Habitats from the Karst Landscape in Tainliao




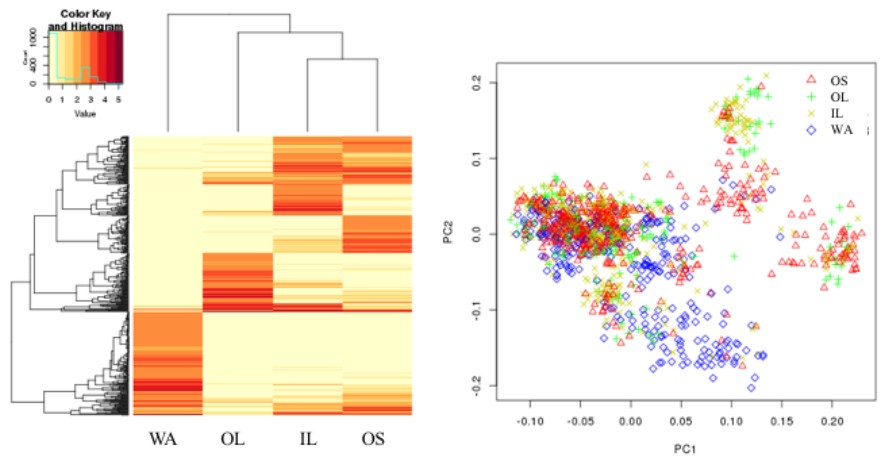

Figure 3. (A) The Heatmap of OTUs Based on the Read Number in Different Habitats in
Tainliao. (B) The PCoA Distribution Based on the Distance Calibration from DNA
Sequences of OTUs in 4 Habitats in Tainliao.




Table 2. The Taxonomy of OTUs and the Bacterial References with the Sequencing
Similarity Higher than 95% to Weathering-Associated Bacteria

| Classification | OTU | Reference of bacteria | Sequence ID | Identities | Taxonomy |
|---|---|---|---|---|---|
| Alphaproteobacteria | karst949 | Labrys sp. | LC372609.1 | 398/407(98%) | Labrys |
| | karst918 | Sphingomonas anadarae | AB261013.1 | 394/405(97%) | Sphingomonas |
| | karst12 | Sphingomonas sp. | AF385529.1 | 400/406(99%) | Sphingomonas |
| | karst1757 | Sphingomonas sp. | AF385529.1 | 393/406(97%) | Sphingomonas |
| | karst2759 | Sphingomonas sanguinis | D13726.1 | 385/403(96%) | Asticcacaulis |
| | karst1653 | Aminobacter sp. | AB905480.1 | 391/408(96%) | Ensifer |
| | karst341 | Aminobacter sp. | FM886907.1 | 401/407(99%) | Ensifer |
| | karst1961 | Rhizobium leguminosarum | D14513.1 | 401/407(99%) | Rhizobium |
| Betaproteobacteria | karst578 | Janthinobacterium sp. | AM071372.1 | 424/433(98%) | Massilia |
| | karst3282 | Janthinobacterium sp. | AB252072.1 | 417/429(97%) | Massilia |
| | karst5 | Collimonas sp. | FR729923.1 | 419/432(97%) | Noviherbaspirillum |
| | karst1267 | Collimonas sp. | FR729923.1 | 413/437(95%) | Ralstonia |
| Gammaproteobacteria | karst1216 | Enterobacter | AB616140.1 | 431/433(99%) | Enterobacter |
| | karst397 | Citrobacter rodentium | AB682287.1 | 415/432(96%) | Escherichia/Shigella |
| | karst3185 | Shewanella morhuae | AB205576.1 | 421/433(97%) | Shewanella |
| | karst595 | Pseudomonas stutzeri | AJ006107.2 | 431/435(99%) | Pseudomonas |
| | karst2678 | Pseudomonas fluorescens | FJ972536.1 | 422/433(97%) | Pseudomonas |
| | karst464 | Pseudomonas sp. | AJ417069.1 | 425/434(98%) | Pseudomonas |
| Gram-positive | karst1814 | Pimelobacter simplex | AY509240.1 | 411/423(97%) | Nocardioides |
| | karst2873 | Arthrobacter oxydans | LN774480.1 | 395/413(96%) | Arthrobacter |
| | karst175 | Streptomyces lividans | AB184695.1 | 403/411(98%) | Streptomyces |
| | karst372 | Mycobacterium colombiense | AM062764.1 | 401/423(95%) | Mycobacterium |
| | karst3247 | Mycobacterium colombiense | AM062764.1 | 407/427(95%) | Mycobacterium |
| | karst1108 | Mycobacterium colombiense | AM062764.1 | 401/424(95%) | Mycobacterium |
| | karst1032 | Mycobacterium colombiense | AM062764.1 | 406/425(96%) | Mycobacterium |
| | karst2144 | Mycobacterium ratisbonense | AJ271863.1 | 393/413(95%) | Mycobacterium |
| | karst101 | Mycobacterium sp. | X84978.1 | 408/411(99%) | Mycobacterium |
| | karst1188 | Bacillus subtilis | AB018487.1 | 417/434(96%) | Bacillus |
| | karst1300 | Bacillus mycoides | AB547222.1 | 432/432(100%) | Bacillus |
| | karst47 | Pimelobacter simplex | AY509240.1 | 399/411(97%) | Nocardioides |
| | karst886 | Streptomyces lividans | AB184826.1 | 399/409(98%) | Streptomyces |
| | karst2729 | Streptomyces lividans | AB184695.1 | 398/413(96%) | Streptomyces |
| | karst2914 | Kocuria polaris | AJ278868.1 | 403/426(95%) | Arthrobacter |



Table 3: The Taxonomy of OTUs and the Bacterial References with the Sequencing
Similarity Higher than 95% to Urease-Producing Bacteria

| Classification | OTU | Reference of bacteria | Sequence ID | Identities | Taxonomy |
|---|---|---|---|---|---|
| Actinobacteria | karst1300 | Bacillus mycoides | AB547222.1 | 432/432(100%) | Bacillus |
| Firmicutes | karst791 | Bacillus megaterium | JX893034.1 | 419/433(97%) | Bacillus |
| | karst260 | Bacillus megaterium | JX893034.1 | 416/431(97%) | Bacillus |
| | karst2293 | Bacillus megaterium | JX893034.1 | 411/430(96%) | Bacillus |
| | karst1188 | Bacillus subtilis | AB018487.1 | 417/434(96%) | Bacillus |
| Gammaproteobacteria | karst189 | Halomonas denitrificans | AM229317.1 | 418/432(97%) | Halomonas |
| | karst2279 | Halomonas denitrificans | AM229317.1 | 414/432(96%) | Halomonas |






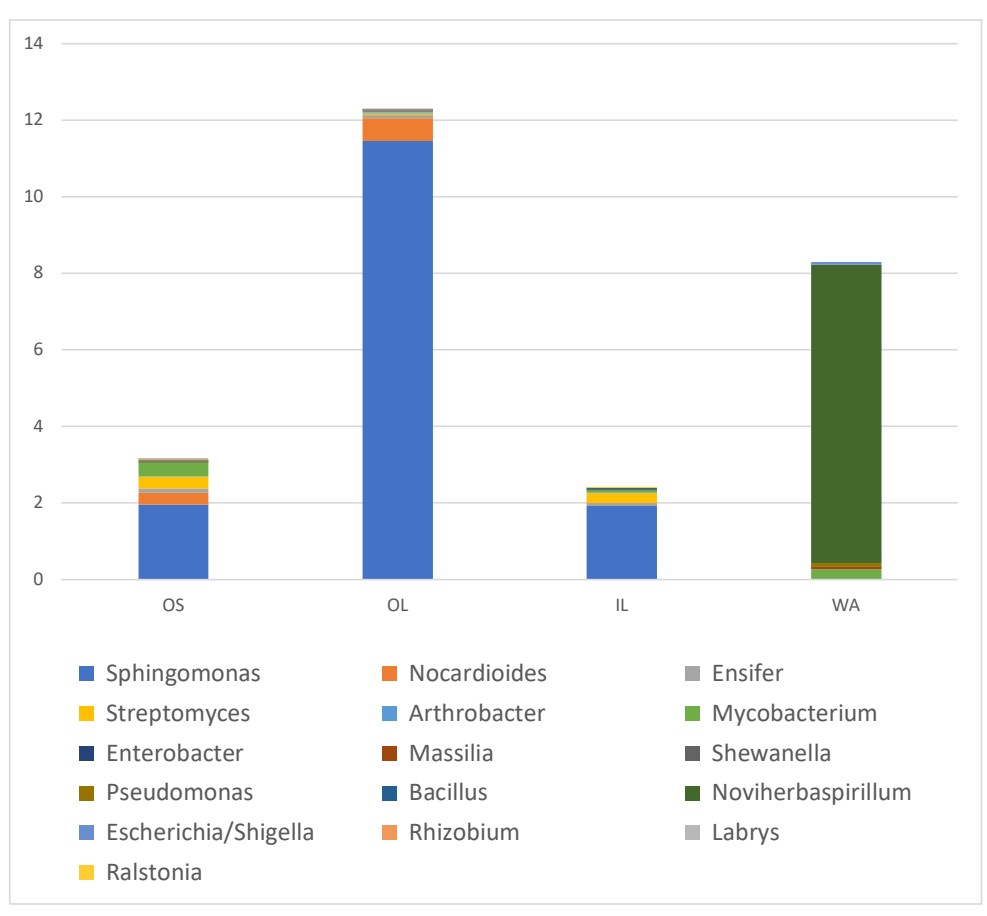


Figure 4: The Relative Abundance of Weathering-Associated Bacterial Lineages in Each
Habitat





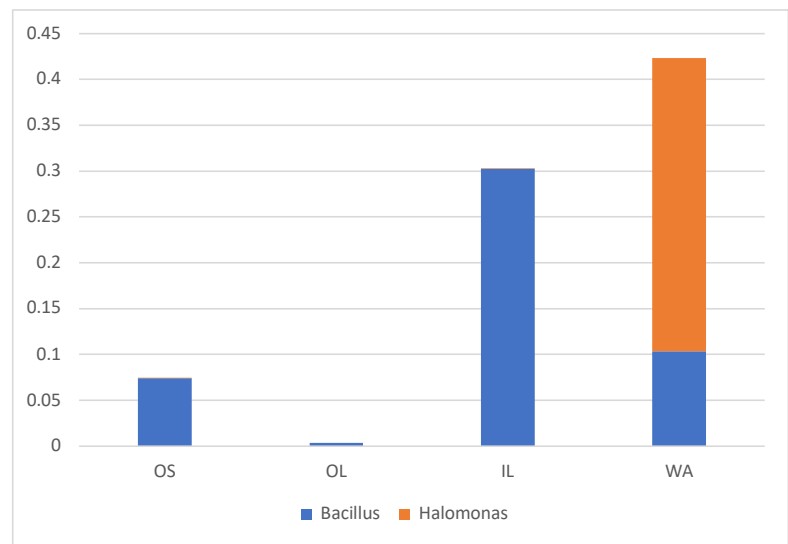


Figure 5: The Relative Abundance of Urease-Producing Bacterial Lineages in Each Habitat



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
