# Peer review of "Role of Microbial Communities in the Weathering and Stalactite Formation in Karst"

_Biogeosciences, 2019_

## Referee Comment (RC1) · Anonymous Referee #1 · 16 Mar 2019

In this study the authors investigated changes in microbial communities across four different sites in a karst landscape, ranging from soils outside a gulch to water dripping from stalactites. They specifically focused on bacteria involved in weathering and urease-activity. These groups were identified based on 16S rRNA gene sequence data by searching for taxa that had been described in previous studies to carry out these functions. They set their findings in relation to different environmental parameters such as light penetration across sites and draw some conclusions regarding the role of bacteria in limestone weathering and calcite precipitation. The authors address the interesting question of how microbes are involved in central geological processes in karstic landscapes. However, I have some concerns regarding the interpretation of

the data obtained in this study. The data set used by the authors is rather limited - as far as I understood, samples were obtained at one time point from four different sites without spatial replicates. In addition, the functional role of the microorganisms is solely inferred from their taxonomic affiliation. Here, at least for the "urea-producing bacteria", additional quantification based on functional genes encoding urease would have been helpful. This would allow comparisons of absolute abundances of this group across the different sampling sites and provide a better approximation of the potential impact of the organisms' activity. It was also a bit surprising here that the authors included a soil sample, as this will for sure provide different microbial communities and higher biomass compared to the rock-associated environments. Consequently, it is not a big surprise that the authors found the highest species diversity in the soil sample, or differences in TOC content between their different sampling sites. Some more explanations would be helpful why the authors investigated "sunlight penetration" as one of the key parameters in this manuscript. Why did they expect this factor to influence microbial communities? More explanation is needed why the authors expected to find an effect of light penetration on the growth of heterotrophic microorganisms (l. 360-362). The conclusions drawn regarding this aspect are very speculative, and the connections between different aspects or parameters are not clear (l. 372-374; l. 379-381; l. 389-390). The description of the scientific question in the introduction remains rather broad (l. 92-93) or refers to aspects which were not addressed in this study (l. 93-95). Similarly, also the objective remains rather unspecific (l. 120-121). Here, some more specific objectives or hypotheses would be valuable. What effect of the tested environmental parameters on microbial communities did the authors expect to find? How does that add to our existing knowledge? The discussion in parts remains very speculative (e. g., l. 324-335; l. 351-354) and the numbers on which discussed differences are based are not always convincing (e. g., l. 333: a factor 2 difference can also be due to variation introduced by the molecular analysis pipeline). In several places, the authors should tone down their conclusions in light of the aspects I listed above. Sequence data originating from four sampling sites and one time point provide only limited support for the two last statements made in the conclusion section (l. 387-390). Here, different experimental approaches would be needed to demonstrate the actual activity of the microorganisms leading to the postulated biogeochemical effects.

Specific comments: l. 81-83: Please add a reference here. l. 89: What is developing orientation? Please explain. l. 105-116: This is a rather detailed description, and parts of this could be moved to the methods section. l. 122 ff: As far as I can see, the authors analyzed data derived from amplicon sequencing in this study. Please avoid the term "genome studies" or "metagenomics" here (and in other places later), because it might be misleading. l. 126-132: This is a very detailed description of the methodological approach which should rather be integrated in the methods section. Please provide information about the key outcomes of your study here instead. l. 136-139: Did the authors take any spatial or temporal replicates? Please explain. l. 168: How did the authors define "weathering-associated bacteria"? Please explain here. l. 178: I wonder if a sequence identity threshold of 95% will be enough to unambiguously identify urease-producing bacteria. Why did the authors not use a different approach by directly targeting functional genes coding for urease? l. 199: What were the exact temperatures here? l. 223-227: This sentence is a bit misleading since the bacterial phyla that are first listed as the four major phyla in all the groups obviously only represent minor fractions in some of the groups. Please rephrase. l. 235-240: This section is not clear, please revise. l. 257-258: How did the authors define "relative impact" here? Is this something that would require activity measurements to be assessed? l. 277-278: which is consistent with its ecological landscape - what does this mean? Please explain. l. 286-287: "geological evolution after their interaction" - what does this mean and was it addressed in this study? l. 290: ...that dominant species in karst samples were affiliated with Actinobacteria and Proteobacteria, please rephrase. l. 293: "extreme diversity" - please provide a comparison to other environments or studies. l. 297-300: This sentence is difficult to understand. Please rephrase. l. 301-303: This sentence is very speculative. l. 305-308: As far as I can see, the authors did not use a metagenomics approach in this study, and the "evolution of microbial communities

and the consequential changes in the environment" have not been studied here in this work. Please rephrase. l. 309, l. 311: How do the authors define "functional bacteria"? Please explain. l. 313-315: Why do the authors address the cutoff question here? What would have been alternative sequence identity cutoffs to use? l. 321-323: This hypothesis does not agree with the objective stated in the introduction. In addition, this seems like a topic that cannot be addressed by a 16S rRNA sequencing approach. Table 1: What is "effective number of species"? Please explain.

———————————————

---

## Referee Comment (RC2) · Anonymous Referee #2 · 4 Apr 2019

The paper 'Role of Microbial Communities in the Weathering and Stalactite Formation in Karst Topography' attempts to connect microbial metabolic activity to both dissolutional and depositional processes with landscape-scale processes in the evolution of karst. Unfortunately, the manuscript suffered from several major problems, the greatest of which was a complete disregard for the vast body of research on hydrogeological processes in shaping karst. This error was compounded by:

1. The very small number of sample sites 2. A lack of cause-and-effect studies to demonstrate a direct role for microorganisms in the describe processes 3. A lack of understanding of general calcification processes (ureolysis is not the sole mechanism

of calcification) 4. No description of the source of urea that could drive the putative calcification processes 5. A 95% identity to a ureolytic species has no bearing on whether the identified phylotype is also ureolytic

I would recommend that the authors work with a geologist and/or geomicrobiologist to better understand the processes they are examining, dramatically expand the sample sites being tested, and develop assays that can directly test whether the microbial activities they are examining are tied to the geologic processes they observe.

---

## Author Comment (AC1) · 25 Apr 2019

In this study the authors investigated changes in microbial communities across four different sites in a karst landscape, ranging from soils outside a gulch to water dripping from stalactites. They specifically focused on bacteria involved in weathering and urease-activity. These groups were identified based on 16S rRNA gene sequence data by searching for taxa that had been described in previous studies to carry out these functions. They set their findings in relation to different environmental parameters such as light penetration across sites and draw some conclusions regarding the

role of bacteria in limestone weathering and calcite precipitation. The authors address the interesting question of how microbes are involved in central geological processes in karstic landscapes. However, I have some concerns regarding the interpretation of the data obtained in this study. The data set used by the authors is rather limited - as far as I understood, samples were obtained at one time point from four different sites without spatial replicates. In addition, the functional role of the microorganisms is solely inferred from their taxonomic affiliation. Here, at least for the "urea-producing bacteria", additional quantification based on functional genes encoding urease would have been helpful. This would allow comparisons of absolute abundances of this group across the different sampling sites and provide a better approximation of the potential impact of the organisms' activity.

Response: We would like to appreciate Reviewer1's insightful comments. In response to reviewer#1's question regarding the sample size and the functional role of urease-producing bacteria, we are currently working on it by replicating 2 batches of samples for testing the urease-genes at different time point. To conduct the spatial and temporal replicates for testing our theory, samples were gathered from randomly selected areas of 15 cm square. In the surveillance of urease-gene in habitats, quantitative PCR are carried out to quantify the yields of the genes at each habitat. Four primers are employed to test the amounts of urease-gene at various habitats namely, ureC-F TGGGCCTTAAAATHCAYGARGAYTGGG; ureC-Fv2 YBGGHYTDAARMTHCAYGAR-GAYTGGG; ureC-R GGTGGTGGCACACCATNANCATRTC; ureC-Rv2 RRTGGTGR-CAVACCATNANCATRTC. As we are testing the optimal condition for measuring the possible candidates of heterogenous urease-genes as well as other photobacteria, we would like to have another 2-month extension in conducting the confirming tests.

It was also a bit surprising here that the authors included a soil sample, as this will for sure provide different microbial communities and higher biomass compared to the rock-associated environments. Consequently, it is not a big surprise that the authors found the highest species diversity in the soil sample, or differences in TOC content

between their different sampling sites. Some more explanations would be helpful why the authors investigated "sunlight penetration" as one of the key parameters in this manuscript. Why did they expect this factor to influence microbial communities? More explanation is needed why the authors expected to find an effect of light penetration on the growth of heterotrophic microorganisms (l. 360-362). The conclusions drawn regarding this aspect are very speculative, and the connections between different aspects or parameters are not clear (l. 372-374; l. 379-381; l. 389-390).

Response: As the soil sample had the same physical parameters in the environments as outer limestone wall did, we decided to include the habitats into our study. This sample will provide a background information on how was bacterial community shifts in response to the effect of sunlight penetration. Up to date, there is little if any research that has explained the role of sunlight on the development of urease-producing bacteria in microbial community. Nevertheless, researches have shown that all the bacteria of high urease activity were isolated from the dark karst cave. As we observed the landscape of the studied site might correlate speleothem formation with sunlight penetration, that area was a sound spot for testing our hypothesis. We agree that solid quantitative data was required before we made the statements. Two batches of samples including urease-producing bacteria and photobacteria were surveyed by qPCR, which would provide solid evidence to address the role of light in these habitats (line 360-362; l. 372-374; l. 379-381; l. 389-390). Our statements will rephrase according to the upcoming results.

The description of the scientific question in the introduction remains rather broad (l. 92-93) or refers to aspects which were not addressed in this study (l. 93-95). Similarly, also the objective remains rather unspecific (l. 120-121). Here, some more specific objectives or hypotheses would be valuable. What effect of the tested environmental parameters on microbial communities did the authors expect to find? How does that add to our existing knowledge? The discussion in parts remains very speculative (e. g., l. 324-335; l. 351-354) and the numbers on which discussed differences are based

are not always convincing (e. g., l. 333: a factor 2 difference can also be due to variation introduced by the molecular analysis pipeline). In several places, the authors should tone down their conclusions in light of the aspects I listed above. Sequence data originating from four sampling sites and one time point provide only limited support for the two last statements made in the conclusion section (l. 387-390). Here, different experimental approaches would be needed to demonstrate the actual activity of the microorganisms leading to the postulated biogeochemical effects.

Response: We agree with the reviewer #1 and decide to modify our manuscript according to the upcoming data, in which emphasizing the effects of bacteria on the shape-change of karst surfaces and exploring the possible causative role of sunlight. In the 92-95 and l120-121, those two sentences will be rephrased to focus on the role of interested bacteria and possible suggested reasons. Again, we will tone down our conclusions and the discussion in line 324-335 and line 387-390 based on the quantitative data of bacteria. We will add the detail information about sampling process in the revised manuscript.

Specific comments:

l. 81-83: Please add a reference here.

Response: The paragraph will be rephrased as following: "The microbial-induced reaction is mainly carried out by urease-producing bacteria in the presence of ammonium ions in the alkaline environment (Dhami et al., 2017)."

Dhami, N. K., Alsubhi, W. R., Watkin, E., and Mukherjee, A.: Bacterial Community Dynamics and Biocement Formation during Stimulation and Augmentation: Implications for Soil Consolidation, Front Microbiol, 8, 1267, 10.3389/fmicb.2017.01267, 2017.

l. 89: What is developing orientation? Please explain.

Response: The paragraph will be rephrased as following: "Temperature, light intensity, and light penetration are important parameters that manipulating the growth of microorganisms. Researchers have shown that microorganisms, operating together with the local environmental conditions, play important roles in remodeling the landscapes of karst (Castanier et al., 1999; Mortensen et al., 2011; Qabany et al., 2012; Anbu et al., 2016)."

l. 105-116: This is a rather detailed description, and parts of this could be moved to the methods section. Response: Thank you for the suggestion, the description will be move to the materials and methods part.

l. 122 ff: As far as I can see, the authors analyzed data derived from amplicon sequencing in this study. Please avoid the term "genome studies" or "metagenomics" here (and in other places later), because it might be misleading.

Response: To avoid confusion, the term "genome studies" or "metagenomics" will be replaced with more descriptive terms according to the needs at different paragraphs.

l. 126-132: This is a very detailed description of the methodological approach which should rather be integrated in the methods section. Please provide information about the key outcomes of your study here instead. Response: The paragraph in line 126-132 will be moved to the methods section. The key outcome together with some highlight of our modification will be presented here.

l. 136-139: Did the authors take any spatial or temporal replicates? Please explain.

Response: Our intention was to propose a novel idea of microbiological roles in the development of karst surface and shape change by using the data from NGS platform. As we considered that an empirical testing on the hypothesis will make this study a valuable contribution to scientific research, we are taking a further step to complete the work. For collecting the samples in the surface of limestone walls and soil, sterile cotton swabs were used to wipe the surfaces of sampled spots, which were randomly selected areas of 15 cm square from each habitat. For the water samples, sterilized bottles (250 ml) were used to collect the water dripping from the stalactite. A total of

4 samples were sent to the laboratory for DNA extraction for total bacterial community prediction in each habitats and quantification of bacteria of interested. The total organic carbon and physical factors in sampling spots including illumination, temperature of the air or soil, humidity, and pH of soil were also measured.

l. 168: How did the authors define "weathering-associated bacteria"? Please explain here.

Response: We will adjust the paragraph to address "weathering-associated bacteria" as suggested. The paragraph will be rephrased as following: "To investigate urease-producing bacteria and weathering-associated bacteria in each habitat, a bioinformatics approach was used to find functional bacteria based on the similarity of DNA sequences. The weathering-associated bacteria were defined as microbial which could facilitate the reaction of mineral dissolution. In this method, . . ."

l. 178: I wonder if a sequence identity threshold of 95% will be enough to unambiguously identify urease-producing bacteria. Why did the authors not use a different approach by directly targeting functional genes coding for urease?

Response: The 95% threshold should be good for the prediction of relative abundance of urease-producing bacteria and give the reasonable basis for this study. We agree with reviewer1 in using a different approach to confirm the quantity of encoded functional genes in each habitat. The empirical test is now undergoing.

l. 199: What were the exact temperatures here?

Response: The paragraph will be rephrased as following: " The temperatures in the air and on the ground of the gulch were (32.4± 1.6 and 28.4±0.1 degree) in inner versus (29.4±1.0 and 26.0 ±0.1 degree) in outer, respectively"

l. 223-227: This sentence is a bit misleading since the bacterial phyla that are first listed as the four major phyla in all the groups obviously only represent minor fractions in some of the groups. Please rephrase.

Response: The paragraph will be rephrased as following: "Proteobacteria was one of the major bacteria that was presented in all the groups. Actinobacteria, which was also a major bacteria in most habitats, contained only limited numbers in water sample. Acidobacteria accounted for marginal portions of relative abundance in samples of inner and outer limestone walls, while Cyanobacteria was present in both limestone walls and was scarcely present if at all in water and soil habitats. Although Actinobacteria can be found in freshwater habitats, our results revealed that they accounted for only <0.4% of the relative abundance in the karst dripping water."

l. 235-240: This section is not clear, please revise.

Response: The Figure 3 will be changed and the paragraph will be rephrased as following: "The square box A in Figure 3B shows the common OTUs that occur in all habitats. However, many OTUs in the water habitat (blue square) were only present in the box of square B (Figure 3B). Figure 3B also showed that the location distribution of OTUs in inner (orange cross) and outer (green cross) karst walls was adjacent to each other. Since the distance in PCoA figure indicated the similarity of DNA sequences between samples, the bacterial DNA in the inner and outer karst wall may similar to each other, suggesting the karst environment may play a decisive role on the growth of bacteria in the two habitats."

l. 257-258: How did the authors define "relative impact" here? Is this something that would require activity measurements to be assessed?

Response: The paragraph will be rephrased as following: "However, the contribution of these bacteria on the weathering process remains unclear."

l. 277-278: which is consistent with its ecological landscape - what does this mean? Please explain.

Response: The paragraph will be rephrased as following: "The surface of the inner wall was covered mostly by stalactite while there were high portions in relative abundance

of urease-producing bacteria on the habitat. This suggests a correlation of stalactite formation the large numbers of urease-producing bacteria may exist."

l. 286-287: "geological evolution after their interaction" - what does this mean and was it addressed in this study?

Response: The paragraph will be rephrased as following: "Understanding the microbial diversity in the karst landscape provides insights into what bacteria survive in extreme environments and how they affect the development of the habitat."

l. 290: ...that dominant species in karst samples were affiliated with Actinobacteria and Proteobacteria, please rephrase.

Response: The paragraph will be rephrased as following: "Most of these studies have confirmed that dominant species in karst samples were affiliated with Actinobacteria and Proteobacteria."

l. 293: "extreme diversity" - please provide a comparison to other environments or studies.

Response: The paragraph will be rephrased as following: " . . . approximately 3500, suggesting the extreme diversity of microorganisms in karst landscapes when compared with other studies (Ortiz et al., 2014, Huang et al., 2018)."

l. 297-300: This sentence is difficult to understand. Please rephrase. Response: The paragraph will be rephrased as following: "Our data of karst soil revealed that this habitat exhibited the highest microbial diversity. Since weathering bacteria present in the outer karst wall and soil were helping the releasing of nutrient and the elevation of biological growth in the soil, a higher total organic carbon content in the soil habitat was expected."

l. 301-303: This sentence is very speculative.

Response: This sentence will be deleted.

l. 305-308: As far as I can see, the authors did not use a metagenomics approach in this study, and the "evolution of microbial communities and the consequential changes in the environment" have not been studied here in this work. Please rephrase.

Response: The paragraph will be rephrased as following: "With the application of the NGS platform on the prediction of relative abundance followed by an empirical quantitative measurement by qPCR, we could examine the effects of physical parameters on the evolution of microbial communities and the consequential changes in the microenvironment."

l. 309, l. 311: How do the authors define "functional bacteria"? Please explain.

Response: Prior to line 309, we added a sentence to define "functional bacteria". The paragraph will be rephrased as following: "Bacteria of the same OTU or with the similar DNA sequence between OTUs may most likely carry the same function in a habitat, which we refer to them as functional bacteria. We used the sequence similarity tool, BLAST, to determine the likeness of representative DNA sequences of OTUs compared with the functional bacteria available in the NCBI database."

l. 313-315: Why do the authors address the cutoff question here? What would have been alternative sequence identity cutoffs to use?

Response: Since we will provide the quantitative data of urease-producing and other bacteria, the cutoff question is no longer important. The paragraph will be deleted.

l. 321-323: This hypothesis does not agree with the objective stated in the introduction. In addition, this seems like a topic that cannot be addressed by a 16S rRNA sequencing approach.

Response: The paragraph will be rephrased as following: "We performed quantitative measurement for urease-producing bacteria and calculated the relative abundances of weathering-associate bacteria in NGS platform."

Table 1: What is "effective number of species"? Please explain.

[Figure]

Response: The paragraph will be rephrased as following: "The symbols of OS, OL, IL, and WA represent sample sites of outer ground, outer limestone wall, inner limestone wall and water, respectively. The effective number of species is equal to the exponential of Shannon-Wiener index."

Please also note the supplement to this comment:
https://www.biogeosciences-discuss.net/bg-2019-12/bg-2019-12-AC1-supplement.pdf

---

## Author Comment (AC2) · 25 Apr 2019

Anonymous Referee #2 The paper 'Role of Microbial Communities in the Weathering and Stalactite Formation in Karst Topography' attempts to connect microbial metabolic activity to both dissolutional and depositional processes with landscape-scale processes in the evolution of karst. Unfortunately, the manuscript suffered from several major problems, the greatest of which was a complete disregard for the vast body of research on hydrogeological processes in shaping karst.

This error was compounded by: 1. The very small number of sample sites

[Figure]

Response: For collecting the samples in the surface of limestone walls and soil, sterile cotton swabs were used to wipe the surfaces of sampled spots, which were randomly selected areas of 15 cm square from each habitat. For the water samples, sterilized bottles (250 ml) were used to collect the water dripping from the stalactite. A total of 4 samples were sent to the laboratory for DNA extraction for total bacterial community prediction in each habitat and quantification of bacteria of interested. Our intention was to propose a novel idea of microbiological role in the development of karst surface and shape change by using the data from NGS platform. As we considered that an empirical testing on the propose will make this study a valuable contribution to scientific research, we are taking further step to complete the work.

2. A lack of cause-and-effect studies to demonstrate a direct role for microorganisms in the describe processes

Response: In response to reviewer#2's question regarding the lack of cause-and-effect studies to demonstrate a direct role for microorganisms in the described processes, we are currently working on it by replicating 2 batches of samples for testing the urease-genes at different time points. To conduct the spatial and temple replicates for testing our theory, samples were gathered from randomly selected areas of 15 cm2 square. In the surveillance of urease-gene in habitat, quantitative PCR is carried out to quantify the yields of the genes at each habitat. As we are testing the optimal condition for measuring the possible candidates of heterogenous urease-genes as well as other photobacteria. We would like to have another 2 month extension in conducting the confirming tests.

3. A lack of understanding of general calcification processes (ureolysis is not the sole mechanism of calcification)

Response: We agree with the comment of reviewer #2 and we will add the description of general calcification processes in the revised manuscript.

4. No description of the source of urea that could drive the putative calcification processes

Response: Previous studies showed that the urease-producing bacteria in the presence of ammonium ions in the alkaline and calcium-rich environment. Our study showed relatively large proportion of urease-producing bacteria in the dark environment. We will use quantitative PCR for verification the metagenomics outcome.

5. A 95% identity to a ureolytic species has no bearing on whether the identified phylotype is also ureolytic. I would recommend that the authors work with a geologist and/or geomicrobiologist to better understand the processes they are examining, dramatically expand the sample sites being tested, and develop assays that can directly test whether the microbial activities they are examining are tied to the geologic processes they observe.

Response: The 95% threshold should be good for the prediction of relative abundance of urease-producing bacteria and give the reasonable basis for this study. We agree with reviewer 2 in using a different approach to confirm the quantity of encoded functional genes in each habitat. The empirical test is now undergoing.

Please also note the supplement to this comment:
https://www.biogeosciences-discuss.net/bg-2019-12/bg-2019-12-AC2-supplement.pdf